# Development of a Low-Molecular-Weight Aβ42 Detection System Using a Enzyme-Linked Peptide Assay

**DOI:** 10.3390/biom11121818

**Published:** 2021-12-02

**Authors:** Sang-Heon Kim, Eun-Hye Lee, Hyung-Ji Kim, A-Ru Kim, Ye-Eun Kim, Jae-Hong Lee, Moon-Young Yoon, Seong-Ho Koh

**Affiliations:** 1Department of Chemistry and Research, Institute of Natural Sciences, Hanyang University, Seoul 04763, Korea; konasi2@naver.com (S.-H.K.); kimr2122@gmail.com (A.-R.K.); 2Departments of Neurology, Hanyang University Guri Hospital, Guri 11923, Korea; gpdmsdl123@hanmail.net (E.-H.L.); yebong0207@daum.net (Y.-E.K.); 3Department of Neurology, University of Ulsan College of Medicine, Asan Medical Center, Seoul 05505, Korea; garailsikzip@gmail.com; 4Department of Translational Medicine, Hanyang University Graduate School of Biomedical Science & Engineering, Seoul 04763, Korea

**Keywords:** Alzheimer’s disease, amyloid-beta 42, phage display, peptide-based assay

## Abstract

Alzheimer’s disease (AD) is a degenerative brain disease that is the most common cause of dementia. The incidence of AD is rapidly rising because of the aging of the world population. Because AD is presently incurable, early diagnosis is very important. The disease is characterized by pathological changes such as deposition of senile plaques and decreased concentration of the amyloid-beta 42 (Aβ42) peptide in the cerebrospinal fluid (CSF). The concentration of Aβ42 in the CSF is a well-studied AD biomarker. The specific peptide probe was screened through four rounds of biopanning, which included the phage display process. The screened peptide showed strong binding affinity in the micromolar range, and the enzyme-linked peptide assay was optimized using the peptide we developed. This diagnostic method showed specificity toward Aβ42 in the presence of other proteins. The peptide-binding site was also estimated using molecular docking analysis. Finally, the diagnostic method we developed could significantly distinguish patients who were classified based on amyloid PET images.

## 1. Introduction

Alzheimer’s disease (AD) is a neurodegenerative disease that causes the most common form of dementia. It leads to memory loss, cognitive and physical impairment, and finally death [1,2]. The incidence of AD is continuously increasing in the United States and the Republic of Korea [3,4,5]. Although many studies have investigated the treatment of AD, it is still deemed an incurable disease. By the time the symptoms of AD manifest, pathological processes—such as amyloid and tau deposition and related neurodegeneration—have already occurred in the brain. Currently available AD drugs address only the symptoms of dementia, not the underlying AD pathologies. Aducanumab (Aduhelm) was approved by the U.S. Food and Drug Administration in June 2021, and several disease-modifying therapies targeting β-amyloid have been studied; however, there is a long way to go to cure AD [6], leaving early diagnosis as the key to effective treatment [7,8].

There are many well-researched biomarkers of common neurodegenerative diseases such as amyloid protein, tau protein, and α-synuclein [9]. The amyloid-beta 42 (Aβ42) peptide is a one of the well-known biomarkers for the diagnosis of AD. It is produced through proteolytic cleavage of the amyloid precursor protein [10,11]. Although the mechanism of action is unclear, the overproduction of Aβ42 has been widely researched in AD diagnosis [12,13]. Abnormally accumulated Aβ42 is prone to aggregating into senile plaques, which are one of the main pathological hallmarks of AD [14,15]. Due to the accumulation of large molecular aggregates in the brain, the concentration of smaller Aβ42 oligomers, which can pass through the brain-cerebrospinal fluid (CSF) barrier, decreases. Reduced Aβ42 concentration in the CSF can be a sign of AD [16,17,18,19,20].

Aβ42 has been detected in numerous experiments, and most of them have used antibodies to detect Aβ42 directly [21,22,23]. Antibodies are the most representative biological molecules used for target probing, and antibody engineering has been also researched even in AD [24]. Because of their high specificity and binding affinity, they have been used in many diagnostic assays [25]. The enzyme-linked immunosorbent assay (ELISA) is the most commonly used diagnostic assay for detecting target molecules using antibodies. ELISA has some advantages, such as having a simple procedure and high specificity and sensitivity. Moreover, it is generally safe and eco-friendly. ELISA also has some disadvantages, such as instability caused by using antibodies, expensive culture media, and the labor and cost involved in antibody preparation [26]. Moreover, the use of antibodies to detect Aβ42 has an obvious disadvantage: because Aβ42 is a very small protein (4 kDa) and antibodies, produced to react with Aβ42 monomers, have much larger sizes, the c-terminus of Aβ42 in oligomeric structures can be hidden for reaction in a sandwich ELISA system [27,28].

To overcome the limitations of using an antibody as a diagnostic probe, a small peptide probe was investigated. Peptide probes have been suggested as alternatives to antibodies in many studies [29,30,31,32]. Small peptide probes can bind to specific targets because of the non-covalent bonds established between peptides and targets, such as hydrogen bonds, ionic bonds, and van der Waals forces. Furthermore, the use of peptide aptamers for the detection of Aβ42 has some advantages, especially for detecting low-molecular-weight Aβ42 oligomers, owing to their small size. Compared to an antibody, a peptide probe has better accessibility and is far less influenced by steric hindrance when it binds to the target [33,34,35,36]. 

Phage display is the most frequently used biopanning method for screening peptide probes. M13 bacteriophages displaying different peptide sequences on their capsids, according to their genotypes, were used [37]. Phage libraries have 10^10^ random sequences, and these libraries are sorted through high-throughput screening to identify the most specific phage for the target. The “stickier” phages—those that bind more strongly to the target—occupy the phage pools via affinity selection, while the non-binding phages are removed through consecutive biopanning rounds. Phage display is a useful technique for researching protein–protein interactions, drug discovery, and estimating the epitope of the antigen. It is especially helpful for finding new ligands, such as enzyme inhibitors, receptor agonists, antagonists, and probes that have a targeting moiety [38,39].

In this study, we mimicked an ELISA system for Aβ42 detection using a small peptide probe to avoid the disadvantages associated with using an antibody as a diagnostic probe in a sandwich ELISA. We screened novel peptide probes using phage display. The concentrations of Aβ42 in the CSF were estimated and cross-checked with the brain amyloid status based on positron emission tomography (PET) imaging.

## 2. Materials and Methods

### 2.1. Materials

Aβ42 (ab120301), Aβ40 (ab120479), and anti-β amyloid antibody (ab2539) were purchased from Abcam (Cambridge, UK). 1,1,1,3,3,3-Hexafluoro-2-propanol was purchased from Sigma-Aldrich (St. Louis, MO, USA). Protein G–coated plates were purchased from Thermo Fisher Scientific (Waltham, MA, USA). Anti-β-amyloid 1–42 (AB5078P) and anti-β-amyloid 1–40 (AB5074P) antibodies were purchased from Merck (Kenilworth, NJ, USA). M13 phages were purchased from New England Biolabs (Ipswich, MA, USA). The studied Aβ42-binding peptide probe (ABPP) sequences were synthesized by AnyGen (Gwangju, Korea). Streptavidin-HRP was obtained from BD Difco Laboratories (Sparks, NV, USA). 3,3′,5,5′-tetramethylbenzidine (TMB) solution was purchased from R&D Systems (Minneapolis, MN, USA). All chemicals were obtained from commercial sources and were of the highest quality available.

### 2.2. Optimization of the Binding Conditions for Phage Display

We added 100 μL of an anti-β-amyloid 1–42 solution (0, 0.5, 1, 2, 4, 6, 8, or 10 μg/mL) and fluorescein isothiocyanate (FITC)-conjugated antibody (2 μg/mL) in phosphate-buffered saline (PBS; pH 7.4) to the protein G–coated opaque plates and incubated these for 2 h. Wells were washed three times with PBST (PBS with 0.05% Tween 20) after discarding the unbound reactants. The fluorescence intensity (excitation wavelength, 495 nm; emission wavelength, 520 nm) was estimated after three washes. Then, 100 μL of anti-β-amyloid 1–42 (8 μg/mL) in PBS was added to each well and incubated for 2 h. After antibody fixation, an excessive concentration of Aβ42 (200 ng/mL) was added to each antibody-coated plate for 2 h. Monomerized amyloid solutions were prepared as previously described [40]. The optimized plates were used for the next screening step.

### 2.3. In Vitro Phage Screening against Amyloid-Beta 42

A random phage library of 10^10^ random sequences was used for the first round of screening. Each round of biopanning was performed under different conditions (Table 1). Protein G-coated plates were coated with Aβ42 after antibody fixation. Phages at a constant concentration were added to the Aβ42-coated wells, and the non-bound phages were washed out using Tris-buffered saline (TBS) with 0.05% Tween 20. The elution buffer containing 0.2 M glycine-HCl and 1 mg/mL bovine serum albumin (BSA; pH 2.2) was administered to elute the bound phages, and the eluate was neutralized using 1 M Tris-HCl (pH 9.1; Figure 1A). Negative selection was performed to reduce the quantity of non-specific phages, by adding the amplified phages to a plate coated only with the antibody for 1 h. Non-bound phages in the supernatant were used for positive selection. The eluted phages were titrated by testing for plaque forming units. For this test, the base of a sterile Petri dish was covered with LB top agar, followed by a mixture of *E. coli* and a dilution of the bacteriophage sample. Growth of *E.coli* would form a lawn of cells in the top agar layer, whereas phage replication would result in the formation of clear zones or ‘plaques’ [41].

### 2.4. Analysis of the Screened Phages

After four rounds of biopanning, each screened phage presenting its own peptide sequence was analyzed. Sixty phage plaque forming units resulting from biopanning were randomly selected, and the chosen phages were infected with *Escherichia coli* strain ER 2738. Afterwards, plasmid DNA coding for the peptide sequences was extracted via ethanolic precipitation. The extracted DNA was sequenced using the following primer: 5′-CCCTCATAGTTAGCGTAACG-3′ (100 pmol, New England Biolabs). Each analyzed phage was amplified by infection with the ER 2738 strain [42].

### 2.5. Characterization of the Binding Properties of Synthetic Peptides

The studied Aβ-binding peptide probe (ABPP) sequences (S E P Q N I W Q Y L R N) were synthesized by AnyGen (Gwangju, Korea). ABPP was conjugated with biotin as a signal reporter, with a purity greater than 95%, as assessed using high-performance liquid chromatography. The binding affinities of peptides were estimated for each Aβ42- and antibody-coated plate. The coating methods followed the protocol described above, except for the use of PBST instead of TBST. After coating the plate with Aβ42, serially diluted peptide was added to the plate for 1 h. After peptide incubation, all wells were washed four times with PBST. Next, the wells were incubated with 100 µL of streptavidin-HRP (1:4000 in PBST) for 1 h. After incubation, the solution in the wells was washed four times with PBST. TMB solution was added to each well for 15 min. Next, 100 µL of 1 M H_2_SO_4_ was added to the wells to stop the reaction, and the absorbance was measured at 450 nm (OD450) using a microplate reader. To determine the detection range, serially diluted Aβ42 was added to the antibody-coated plates. After washing, the plate was incubated for 1 h with 100 µL of 100 µM peptide. Streptavidin and TMB treatment steps were performed as described above. Based on the measurements, the limit of detection (LOD) for the diagnostic system was calculated according to the formula from the IUPAC: LOD = 3 × *SD*/slope. To verify the specificity of our diagnostic system, other amyloid proteins were added to the antibody-coated plates. Then, 100 µL of 100 µM of peptide was added to each protein-treated plate and incubated for 1 h. Streptavidin and TMB treatment steps were performed as described above.

### 2.6. Estimation of the Binding Site of the Developed Peptides

Anti-Aβ42 and anti-Aβ40 antibodies were applied to the protein G-coated plates and incubated for 2 h. After washing, Aβ42 and Aβ40 were used to consecutively treat each antibody-coated plate for 2 h. Next, 100 µL of 100 µM peptide was added to each protein-treated plate, which was then incubated for 1 h. Streptavidin and TMB treatment steps and absorbance measurements were performed as described above. In addition, two kinds of antibodies—which recognize different epitopes (anti-Aβ40; C-terminus of Aβ40 and anti-amyloid β protein; 1–14 sequence of Aβ40)—were added to the protein G-coated plates and incubated for 2 h. The coated plates were washed thrice with PBST. After antibody coating, the same concentration of Aβ40 was administered to the plates, followed by incubation for 2 h. The plates were cleaned three times with PBST. Then, 100 µL of 100 µM peptide was added to each protein-treated plate and incubated for 1 h. Streptavidin and TMB treatment steps were performed as described above.

### 2.7. Molecular Docking of Aβ42 and Peptide

Molecular docking analysis of the peptide against Aβ42 was performed using AutoDock Vina [43]. The binding site of the peptide to Aβ42 was also estimated. Accordingly, for the Aβ42 crystal structure, we selected the receptor protein 1Z0Q [44]. To calculate a more realistic ligand–protein interaction, the flexible and non-flexible residues were identified and appropriate charges were added. The peptide ligand was modeled with free torsional bonds in its structure. The docking area was calculated using a three-dimensional grid box with grid points of 62 × 60 × 62 Å and a line spacing of 0.375 Å.

### 2.8. Recovery Test of Aβ42 at Different Spiking Levels

PBS and human serum were used as solutions for the spiking test. Each of the Aβ42 and Aβ40 stock solutions was diluted using PBS, and the mixture was diluted using human serum. The diluted samples were added to antibody-coated plates and incubated for 2 h. The binding conditions used for the antibody coating and the peptide, streptavidin, and TMB treatment steps are described above. To calculate the recovery ratio, we divided the signals of the mixtures by the signals of Aβ42.

### 2.9. Participants

This study consecutively and prospectively enrolled 98 participants who visited the memory clinic of Asan Medical Center, Seoul, South Korea, from June 2018 to July 2020. All participants or their proxies provided informed consent. All participants underwent brain MRI, detailed neuropsychological testing, fluorine-18 (^18^F)-florbetaben amyloid PET, and CSF analysis. The participants who were included in the study met the following criteria: they (1) were aged over 40 and under 90 years; (2) showed cognitive impairment, which included subjective cognitive decline (SCD), mild cognitive impairment (MCI), and dementia, and was defined by subject memory impairment reported by the patients or caregivers, activities of daily living, as judged by a physician, and objective memory decline below the 16th percentile (−1 standard deviation) for the demographically matched norms determined via neuropsychological testing; and (3) showed no evidence of structural lesion on a brain MRI that could influence cognitive function. All PET images were obtained using Discovery 690, 710, and 690 Elite PET/computed tomography scanners (GE Healthcare, Chicago, IL, USA). Amyloid PET images were acquired for 20 min, beginning 90 min after injection of 300 ± 30 MBq of ^18^F-florbetaben. Two neurologists (H.J.K. and J.H.L) and two nuclear medicine physicians reviewed the PET scans according to the predefined regional cortical tracer binding and brain amyloid plaque load (BAPL) scoring system [45]. The final score was reached by consensus, with a BAPL score of 1 regarded as Aβ−, and BAPL scores of 2 and 3 considered Aβ+ [46,47]. All participants underwent blood tests, in which a complete blood count, lipid profiles, erythrocyte sedimentation rate (ESR), vitamin B12, folate, homocysteine serum levels, and thyroid function were assessed. The *ApoE* genotype was identified after extracting genomic DNA from the venous blood.

### 2.10. Neuropsychological Assessment

When referring to a pathological change in the AD continuum, it is necessary to consider the clinical status of the patient. All patients were assessed using the Seoul Neuropsychological Screening Battery (SNSB) as a formal test to accurately describe their cognitive status as SCD, MCI, or dementia [48]. The SNSB is a detailed neuropsychological battery that includes various tests that measure attention (forward/backward digit span), language (comprehension, repetition, confrontational naming, reading, and writing), calculation, praxis (buccofacial and ideomotor), visuospatial function (Rey Complex Figure Test; RCFT), verbal memory (Seoul Verbal Learning Test with immediate recall, delayed recall, and recognition), visual memory (RCFT with immediate recall, delayed recall, and recognition), and frontal/executive functions (contrasting program, go/no-go test, verbal fluency, and the Stroop test). We also performed other clinical and cognitive performance measurements, including the Korean version of the Mini-Mental State Examination (K-MMSE), Global Deterioration Scale (GDS), Clinical Dementia Rating (CDR), Korean version of the Neuropsychiatric Inventory, and the 30-item Geriatric Depression Scale (GDS-30).

### 2.11. ELISA

The concentration of Aβ42 in the CSF was examined using a commercial ELISA kit (R&D Systems, Minneapolis, USA) according to the manufacturer’s instructions. Briefly, the CSF samples were added to human Aβ42 monoclonal antibody-coated microplates and incubated for 2 h at 2–8 °C. The unbound substances were washed away, and the molecules presenting only Aβ42 were attached to the capture antibody. Then, the captured Aβ42 was detected using enzyme-linked monoclonal antibodies for another 2 h at 2–8 °C. Substrates were added to the microplates, and color development was measured using a microplate reader at a wavelength of 450 nm.

### 2.12. Verification of the Diagnostic Ability of the Peptide-Based Detection System

CSF samples from patients were added to antibody-coated plates and incubated for 2 h. The binding conditions used for antibody coating and the peptide, streptavidin, and TMB treatment steps are described above.

### 2.13. Statistical Analysis

GraphPad Prism 6 was used for statistical analysis. Statistical analyses of the demographics of the participants were conducted as follows. Student’s *t*-test was used to analyze the age, educational level, disease duration, MMSE, CDR, neutrophil, monocyte, ESR, C-reactive proteins (CRP), and low-density lipoprotein (LDL) levels. The χ^2^ test was used to analyze the sex distribution, apolipoprotein E (*ApoE*) ε4 allele, diabetes mellitus (DM), hypertension (HTN), and hyperlipidemia. For the ELISA and ABPP sensitivity and specificity tests, Student’s *t*-tests and ROC analyses were used.

## 3. Results

### 3.1. Optimization of the Binding Conditions of the Diagnostic System

As the concentration of anti-Aβ42 antibody increased, the fluorescence intensity of the FITC-labeled antibody decreased. Due to occupation by a non-labeled anti-Aβ42 antibody, FITC-labeled antibody could not bind to the plate. Saturation of the anti-Aβ42 antibody was observed at 8 μg/mL (Figure 1B).

### 3.2. Characterization of the Peptide Using the Diagnostic System

The screened peptide sequences are presented in Table 2. Only two different sequences were found at a higher frequency than the other sequences. One peptide sequence (S E P Q N I W Q Y L R N) was chosen for the synthesis. The binding affinities of the screened peptides were calculated. Serially diluted peptides showed saturation according to Hill-slope fitting. The binding affinity of the peptide is shown in Figure 2. The absorbance increased with increasing peptide concentration and reached saturation near a peptide concentration of 100 μM. The dissociation constant (K_D_) for the ABPP was 17.47 ± 2.98 μM (Figure 2A). The detection range of the diagnostic system using our screened peptides was also estimated using serially diluted Aβ42. It showed good linear calibration between the concentration of Aβ42 and absorbance in the range 1.25–80 ng/mL (Figure 2B). In addition, we verified the ability of our diagnostic methods to distinguish between Aβ42 and other proteins in solution. Significantly higher signals were obtained for Aβ42 than for the other proteins (Figure 2C).

### 3.3. Estimation of the Epitope Recognized by the Screened Peptide

Adding Aβ42 to anti-Aβ42 antibody-coated plates presented a large signal contrast compared to the addition of Aβ42 to anti-Aβ40 antibody (Figure 3A,B). The signals obtained by adding Aβ42 to an anti-Aβ42 antibody-coated plate were much higher than those obtained by adding Aβ40 to an anti-Aβ40 antibody-coated plate (Figure 3C,D). When Aβ40 was added, we measured much higher absorbance values with an antibody that could recognize the 1–14 sequence of Aβ40 than when we used an antibody that could recognize seven amino acids from the C-terminus of Aβ40 (Figure 4A). To estimate the peptide-binding site, molecular docking of the screened peptide with Aβ42 was performed. Docking analysis showed that the peptide was bound to the mid-to-end region of the Aβ42 sequence. In particular, the polypeptide chains of K16 and F20 in Aβ42 formed hydrogen bonds with the peptide chains of R11 and L10 in the screened peptide. In addition, hydrogen bonds between the peptide chains of the 30A, I31, I32, G33, and L34 residues in Aβ42 and the peptide chains of the S1, E2, and W7 residues in the peptide ligand were formed. The binding energy between ABPP and Aβ42 was −4.3 kcal/mol (Figure 4B).

### 3.4. Recovery Test of Aβ42

Different concentrations of Aβ42 and Aβ40 were spiked into blank PBS and human serum samples. They were analyzed using the methods described above. In all spiked mixtures, the average recoveries of Aβ42 ranged from 96.03% to 104.51% and showed a %RSD of approximately 10% when using PBS as the solution (Figure 5 and Table 3).

### 3.5. Characteristics of Participants

The detailed demographic and clinical characteristics of the participants are presented in Table 4. The members of the Aβ− group were older than those in the Aβ+ group. The Aβ+ group included 24 patients with early onset AD. Except for the early onset subjects, there was no difference in age between the two groups. Of note, the frequency of the *ApoE* ε4 allele was significantly higher in the Aβ+ group (76.9%) than in the Aβ− group (23.1%). The frequency of DM was higher in the Aβ− group than in the Aβ+ group. Serum ESR was higher in the Aβ− group, and LDL levels were higher in the Aβ+ group. Other demographic factors, including sex distribution, disease duration, educational level, frequency of HTN, and hyperlipidemia, were not significantly different between the two groups.

### 3.6. Accuracy Validation of the Diagnostic Methods by Cross Checking with the PET Images

Amyloid PET imaging is one of the main medical approaches used to diagnose AD. The participants were divided into two groups—amyloid-positive and amyloid-negative—based on the existence of deposited amyloid proteins (Figure 6). ELISA and the diagnostic system using ABPP were performed on the CSF samples of the participants, and their specificities and sensitivities regarding Aβ42 detection were compared. The amyloid-positive group showed significantly lower levels of Aβ42 in the CSF than the amyloid-negative group, according to both detection methods (Figure 7A,B). ELISA classified the amyloid PET-positive and -negative participants with 67.3% sensitivity and 69.6% specificity, whereas the peptide system had 61.5% sensitivity and 65.2% specificity (Figure 7C,D).

## 4. Discussion

In this study, we developed an Aβ42 detection system using a peptide as a signaling probe. Although sandwich ELISA is an easily accessible method for measuring target concentrations, it is not very precise when detecting Aβs. To overcome the limitations of the general sandwich ELISA, which could not be used to analyze the lower molecular weight amyloid aggregates, we attempted to develop a peptide probe for Aβ42. Unlike sandwich ELISA, our developed method uses antibodies only once to capture the target.

The peptide was screened using phage display as a signaling probe. For biopanning, Aβ42 was fixed onto the plate and captured by the antibody. Then, phage binding, washing, elution, and amplification for the next screening rounds proceeded under harsh conditions. Due to the use of an antibody as a capture probe, our system shows specificity toward Aβ42. For the fixation of antibodies on the plate surface, we used protein G-coated plates, which increased the efficiency of antibody fixation by helping the antibodies face the same direction [49]. Fluorescence was measured to optimize the antibody-binding conditions. The fluorescence intensity decreased with increasing concentrations of the anti-Aβ42 antibodies. Decreased fluorescence intensity indicates that the non-labeled anti-Aβ42 antibody was packed on the surface of the plate and saturated in the same direction.

Among the screened peptides, we selected one sequence based on its solubility and synthesized it. Another sequence had too few hydrophilic residues, which affected its solubility. Because hydrophobic sequences require organic solvents, we synthesized the most hydrophilic sequence [50]. The synthesized peptide was labeled with biotin for signaling, and the absorbance increased as the concentration of the peptide increased. The binding affinity of the biotin-labeled peptide was estimated to be in the micromolar range. Due to its strong affinity, ABPP was used as a signaling probe for the next procedure.

To quantify Aβ42, we performed regression analysis. The high *R*^2^ value indicates that our standard curve showed a good linear relationship between the OD450 value and Aβ42 concentration [51]. For accuracy, a diagnostic method was needed that emits signals only in response to Aβ42. Four prion proteins were used to validate specific binding during the analysis. Among them, Aβ42 showed much higher absorbance values than the others, demonstrating that our analysis method can distinguish Aβ42 from other proteins in solution. To verify the specificity of our analysis method, we used two types of antibodies that can target either Aβ42 or Aβ40. Much lower absorbance values were obtained when using the anti-Aβ40 antibody than when using the anti-Aβ42 antibody, which indicates that the specificity derives from the use of a suitable antibody. Furthermore, because of the consensus sequences between Aβ proteins, the specificity of ABPP was estimated by adding Aβ proteins to anti-Aβ antibody-coated plates. The absorbance signals observed when Aβ40 was added to anti-Aβ40 antibody, which recognizes the C-terminus of Aβ40 (from L34 to V40), were completely different from the absorbance signals observed when Aβ42 was added to anti-Aβ42, despite their consensus sequences. Interestingly, however, the absorbance resulting from adding Aβ40 to antibody-coated plates that recognized different epitopes (from D1 to H14) showed almost the same signals as the established Aβ42 detection. It seems that the recognition site of ABPP in Aβ was affected and overlapped with the epitope of the anti-Aβ40 antibody (from L34 to V40). To predict noncovalent interactions between peptide and Aβ42, molecular docking analysis was performed, and the results also supported the comparison results using antibodies. The seven amino acid residues of Aβ42 were found to interact with five residues of ABPP forming hydrogen bonds. These amino acid residues of Aβ42 that are involved in peptide binding overlapped partially with the epitope of the anti-Aβ40 antibody (from L34 to V40), and most of them were next to the epitope that can be affected by steric hindrance [52]. To verify the ability to detect Aβ42 not only from Aβ42 dilutions, but also from the protein mixture, Aβ42 was spiked and recovered. The high average percentage recovery means that our diagnostic methods can segregate the target, even if it is in a protein mixture.

Fluorine-18–florbetaben amyloid PET is a useful clinical tool for the visualization of amyloid plaques in patients with AD [53]. The PET image showed that uptake of the PET tracer diminished the demarcation of the gray and white matter. This means that ^18^F-florbetaben detected cortical amyloid aggregates in patients. All participants were divided into two groups based on their amyloid PET results: amyloid-positive and -negative. Our dataset included 24 subjects aged <65 years who had an early onset of symptoms. Therefore, the apparent difference in age and baseline cognitive function between the two groups can be attributed to these early onset patients. The higher number of *ApoE* ε4 allele carriers in the amyloid-positive group was also consistent with that in previous studies [54]. Furthermore, PET results—amyloid-positive or -negative—can be used as a criterion for the precision of ELISA and the diagnostic methods we developed. Interestingly, although there was also an overlapping area, the absorbance estimated using ABPP-based diagnostic methods showed a very significant difference between the amyloid-positive and -negative groups. In the amyloid-positive group, the absorbance values were estimated to be lower than those in the amyloid-negative group, similar to the ELISA results. The mean concentration of Aβ42 in the CSF of AD and non-AD patients was 410 ± 200 pg/mL and 657 ± 208 pg/mL, respectively [55]. As described above, the concentration of Aβ42 in the CSF decreases due to the deposition of Aβ42 in the brain, which is consistent with our results [56]. This means that our screened ABPP and the developed method can distinguish patients who have Aβ42 depositions in the brain and can be helpful for Aβ42 quantification, even in the monomeric state. Aprile et al. precisely showed that their newly developed single-domain antibody specifically recognized the oligomeric form of Aβ42 rather than the other forms such as monomeric and fibrillar [24]. However, the recognition of the monomeric form of Aβ42 by the antibody was not mentioned therein. As shown in our supplementary results (Appendix A), our peptide can bind to monomeric as well as some oligomeric forms of Aβ42, thus, the peptide could be used to detect these forms of Aβ42. It would have been ideal if we tested the binding of our peptide to different conformers rather than doing TEM, which is subjective to surface-mass transport issues during sample deposition and differential binding of Aβ42 conformers to the TEM grid. TEM alone might not add much evidence as to which species our peptide binds. In addition, our results would be strengthened if we could quantify the binding of our peptide to at least monomers and fibrils (and ideally also stabilized oligomers). Therefore, we think further studies might need to be done to deal with these issues in the future. Nonetheless, our research will be helpful for clinical research on AD because of the difficulties in analyzing the relationship between the monomer state of Aβ42 and the progression of AD. Moreover, they can also specifically detect Aβ40 when an appropriate antibody is used for coating.

## 5. Conclusions

In this study, we identified the peptide sequence that bound most strongly to the monomeric state of Aβ42, and our screened peptide showed specific binding to only Aβ42 using the developed system. The binding sites were also estimated and matched using molecular docking analyses. Finally, patients with amyloid depositions in the brain were distinguished using our peptide-based diagnostic method. Our diagnostic method still has problems, such as its sensitivity and specificity; however, our peptide and the diagnostic method are helpful for several clinical research areas in the AD field because of their potential to detect the monomeric state of Aβ42. Using a peptide has several benefits—such as its small size, non-immunogenicity, stability, and ease of modification—not only in terms of research but also in commercial applications. These advantages of using a peptide probe as an alternative to antibodies appear to be relevant to AD research.

## Figures and Tables

**Figure 1 biomolecules-11-01818-f001:**
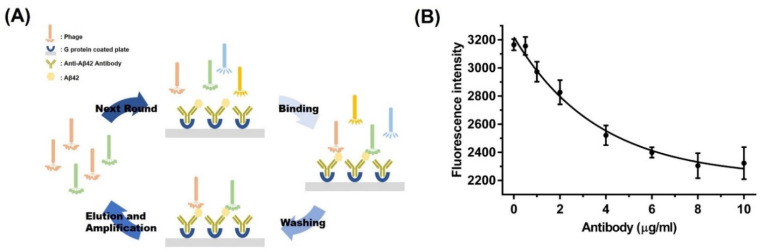
Scheme of biopanning and optimization of the screening conditions. (**A**) Most specific phages were screened via binding, washing, elution, and amplification steps, a procedure called biopanning. (**B**) The fluorescence signal from the FITC-labeled antibodies decreased as the concentration of anti-Aβ42 antibodies increased. The test was performed in triplicate under the same conditions.

**Figure 2 biomolecules-11-01818-f002:**
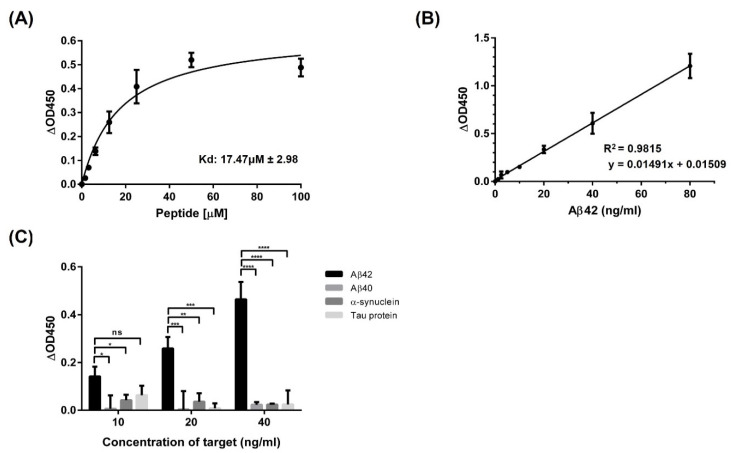
Characterization of binding of the screened peptide. (**A**) The peptide showed strong binding affinity in the micromolar range with a K_D_ of 17.47 ± 2.98 μM. (**B**) Linear regression of binding between Aβ42 and ABPP (*R*^2^ = 0.9815). (**C**) Specificity test on an anti-Aβ42 antibody–coated plate. Only Aβ42 showed meaningful signals. One-way ANOVA analysis was performed for statistical analysis (ns *p* > 0.05, * *p* ≤ 0.05, ** *p* ≤ 0.01, *** *p* ≤ 0.001, **** *p* ≤ 0.0001). All tests were performed in triplicate under the same conditions.

**Figure 3 biomolecules-11-01818-f003:**
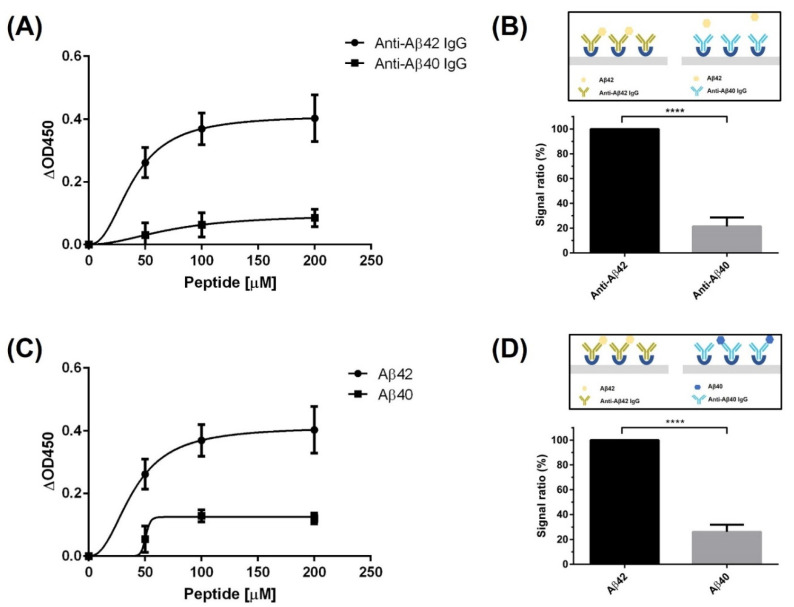
Verification of the Aβ-specific binding in the peptide-based diagnostic method. (**A**) Comparison of signals using two kinds of Aβ-specific antibodies. The signal was shown only when using the anti-Aβ42 antibody when Aβ42 was added to the antibody-coated plate. (**B**) The ratio of signals resulting from using the anti-Aβ42 antibody and the anti-Aβ40 antibody with Aβ42. An unpaired *t*-test was performed for statistical analysis (**** *p* ≤ 0.0001). (**C**) Comparison of signals when Aβ42 and Aβ40 were added to plates coated with antibodies specific to each respective Aβ species. (**D**) The ratio of signals resulting from Aβ42 detection by anti-Aβ42 antibody to those resulting from Aβ40 detection by anti-Aβ40 antibody on antibody-coated plates specific to each of the Aβ species. An unpaired *t*-test was performed for statistical analysis (**** *p* ≤ 0.0001). All tests were performed in triplicate under the same conditions.

**Figure 4 biomolecules-11-01818-f004:**
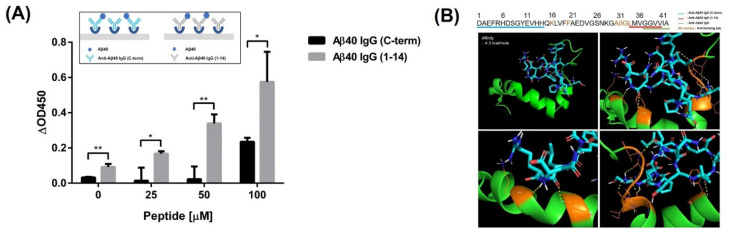
Estimation of the peptide binding sites in Aβ42. (**A**) Signal difference of Aβ40 between two types of antibodies that recognized different epitopes. Multiple *t*-test was performed for statistical analysis (* *p* ≤ 0.05, ** *p* ≤ 0.01). The test was performed in triplicate under the same conditions. (**B**) Molecular docking analysis of peptide binding to Aβ42.

**Figure 5 biomolecules-11-01818-f005:**
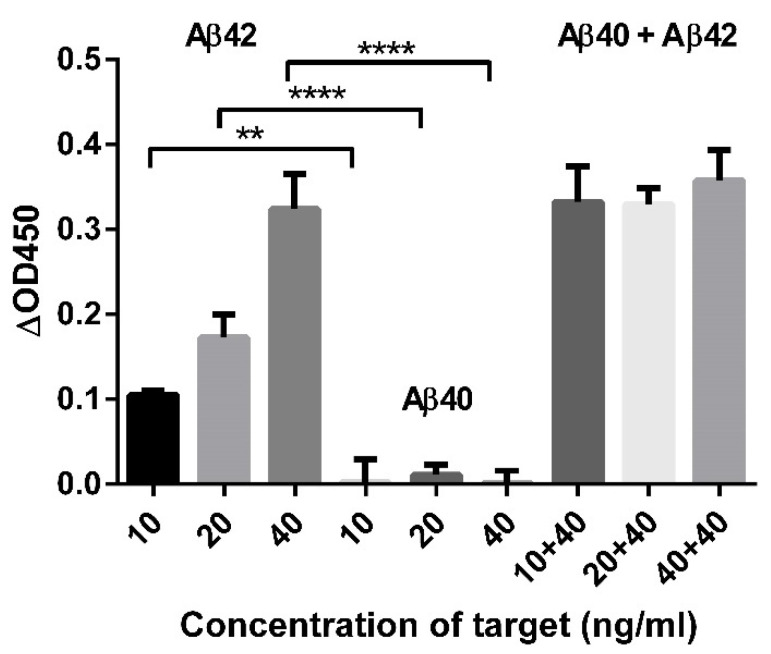
Spiking and recovery in protein mixtures. A specific concentration of Aβ42 was spiked, and the average recovery signals were estimated. One-way ANOVA followed by Dunnett’s multiple comparisons tests were performed for statistical analysis (** *p* ≤ 0.01, **** *p* ≤ 0.0001). All tests were performed in triplicate under the same conditions.

**Figure 6 biomolecules-11-01818-f006:**
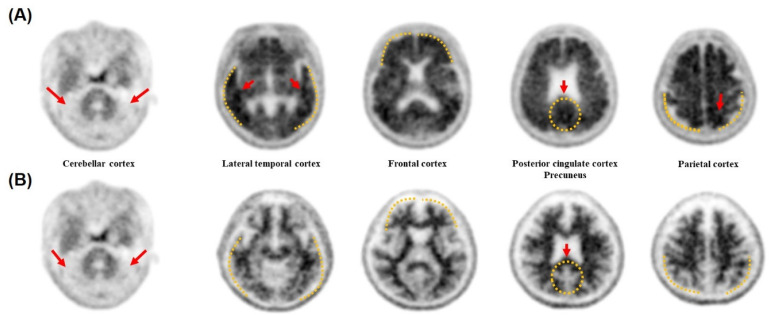
Gray-scale images of fluorine-18 [^18^F]-florbetaben amyloid PET. (**A**) Typical amyloid PET image of BAPL3. An increased florbetaben tracer uptake was observed not only in the bilateral frontal and parietal cortex, but also in the anterior striatum. The differentiation of the cortical gray matter and subcortical white matter disappeared. (**B**) Typical amyloid PET image of BAPL1. The cortical–subcortical demarcation was well maintained, and no florbetaben tracer uptake was observed in the cerebral cortex.

**Figure 7 biomolecules-11-01818-f007:**
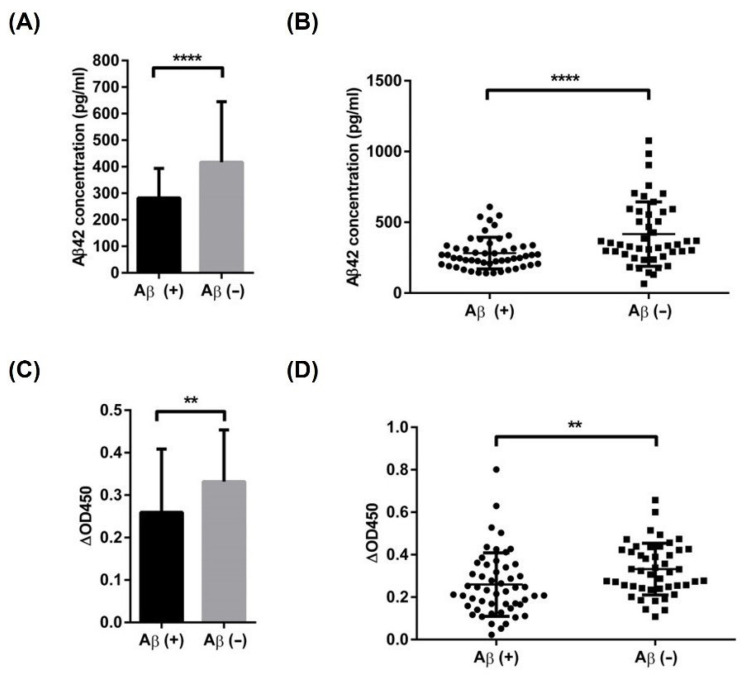
Comparison of the ELISA and peptide systems using the CSF samples. (**A**) The average of the Aβ42 signals estimated using ELISA and the results of the unpaired *t*-test comparing the positive and negative categories (**** *p* ≤ 0.0001). (**B**) Box plot of the Aβ42 signal distributions in each group using ELISA (**** *p* ≤ 0.0001). (**C**) The average of the Aβ42 signals estimated using the ABPP-based method and the results of the unpaired *t*-test comparing the positive and negative categories (** *p* ≤ 0.01). (**D**) Box plot of the Aβ42 signal distributions in each group using the ABPP-based method (** *p* ≤ 0.01).

**Table 1 biomolecules-11-01818-t001:** Conditions of phage display.

Round #	Binding Condition	Binding Time (min)	Washing Conditions
1R	TBS	120	0.1% TBST
2R	TBS	60	0.1% TBST
3R	TBS	60	0.1% TBST
Negative	TBS + 0.5% BSA	60	
4R	TBS + 0.5% BSA	30	0.1% TBST, 500 mM NaCl

**Table 2 biomolecules-11-01818-t002:** Analyzed peptide sequences.

Phage #	Sequence	Frequency (30/60)
9	S E P Q N I W Q Y L R N	20
21	I W M T R T N L N D V N	10

**Table 3 biomolecules-11-01818-t003:** Spiking and recovery rate.

Concentration of Target (Aβ40 + Aβ42, ng/mL)	Average Recovery (%)	Standard Deviation (%)	%RSD
10 + 40	97.02	12.46	12.84
20 + 40	96.04	5.81	6.053
40 + 40	104.5	10.51	10.15

**Table 4 biomolecules-11-01818-t004:** Demographics and characteristics.

		Amyloid-Negative	Amyloid-Positive
Population	Sex, *n* (female)	46 (21)	52 (32)
	Age (SD), y *	71.67 (8.57)	66.73 (10.36)
	Education level (SD), y	11.67 (4.68)	11.41 (4.77)
	Disease duration (SD), m	28.52 (28.89)	35.62 (27.10)
	*ApoE* ε4 carrier, *n* (%) **	9 (23.1)	30 (76.9)
Underlying disease	DM, *n* (%) *	22 (46.9)	10 (19.2)
	HTN, *n* (%)	21 (45.7)	25 (54.3)
	Hyperlipidemia, *n* (%)	18 (40.9)	26 (59.1)
Global cognition	MMSE (SD)	23.91 (4.70)	22.00 (4.90)
	CDR (SD)	0.69 (0.52)	0.72 (0.45)
Laboratory test	Neutrophil, % (SD)	56.42 (9.09)	56.64 (8.91)
	Monocyte, % (SD)	8.15 (1.64)	11.05 (19.01)
	ESR (SD), mm/h *	15.29 (11.65)	10.91 (7.00)
	CRP (SD)	0.23 (0.46)	0.21 (0.48)
	LDL (SD), mg/dL *	104.68 (39.24)	127.03 (38.21)

Student’s *t*-test was used for the analyses of the age, educational level, disease duration, MMSE, CDR, and neutrophil, monocyte, ESR, CRP, and LDL levels. The χ^2^ test was used for the analyses of sex distribution, *ApoE* ε4, DM, HTN, and hyperlipidemia. Abbreviations: DM, diabetes mellitus; HTN, hypertension; MMSE, Mini-Mental Status Examination; CDR, Clinical Dementia Rating; ESR, erythrocyte sedimentation rate; CRP, C-reactive proteins; LDL, low-density lipoprotein; SD, standard deviation. * Significant at *p* < 0.05. ** Significant at *p* < 0.001.

## Data Availability

Due to privacy and ethical concerns, the data that support the findings of this study are available on request from the first authors (S.H. Kim, E.H. Lee, and H.J. Kim).

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
