# Peer review of "Development of a Low-Molecular-Weight Aβ42 Detection System Using a Enzyme-Linked Peptide Assay"

_biomolecules, 2021, doi:10.3390/biom11121818_

Round 1

Reviewer 1 Report

My major comment is that for diagnostic of AD, the use of monomeric Abeta42 is very controversial and has been shown not to follow cognitive impairment in patients. Abeta42 may largely vary in individuals. Therefore, the statement that the proposed approach can be used to diagnose AD is not correct.

2. Incorrect statement: Moreover, the use of antibodies to detect Aβ42 has an obvious disadvantage: because Aβ42 is a very small protein (4 kDa), a much larger antibody can detect only an Aβ42 oligomer over a certain size in a sandwich ELISA system [25, 26]. -antibodies are produced to react with Aβ42 monomers, while the c-terminus of in oligomeric structure can be hidden for reaction.

3. Not clear which sequence of Aβ42 is recognized. Please, also specify the peptide sequence encoded.

4. in Material and Methods, the authors mention that sixty plaques resulting from biopanning were randomly selected; by definition, a plaque is an amyloid formation in the brain tissue. Please explain how it can refer to the in vitro experiment.

5. The authors mentioned that SEPQNIWQYLRN was used to bind Aβ42; please specify if that is specific to Aβ42 c-terminus.

6. Molecular docking of Aβ42 and peptide - should be better explained, it is not clear what authors mean by Flexible and non-flexible residues were identified, and appropriate charges were added. 

7. In the discussion, the statement that Molecular docking analysis 

revealed its mode of action should be better explained.

7. Molecular mechanism of specific 42 detection should be discussed.

8. Instead of multiple t-tests, one-way ANOVA could be used.

Reviewer 2 Report

This is a well written and interesting study from experienced authors. The impact is somewhat limited given issues with specificity and sensitivity, as their construct performs worse than currently available antibodies. Nonetheless, this is expected given they are using a small peptide, and their findings will hopefully serve as a foundation for the development of better binders in the future.

Major concern:

1. Given the claims, it is important to test the binding of their peptide to the various conformers of Ab42, including a purely monomeric solution, stabilized oligomers (such as using the robust ADDL protocol that generates monomers and oligomers), and fibrils. Ideally, this would also be carried out for Ab40.

Minor issues:

1. title: "peptide" is repeated twice in close succession

2. affiliation: recommend updating "all authors" to "these authors... equally..."

3. Line 19: update to "presently incurable"

4. Lines 44 to 52: tau should be discussed given its importance to AD

5. Line 54: F.A. Aprile, PNAS, 2020 is an important reference in this context. Recommend the authors also compare their results to this paper in the discussion. They have done an appropriate job listing the limitations of their construct, but it would be highly helpful to do so more quantitatively. 

6. Line 151: for how long was the peptide incubated? Does this corresponding to a purely monomeric solution? 

Round 2

Reviewer 1 Report

all my questions have been replied.  I have no more comments.

Author Response

Thanks for the reviewer's kind decision.

Reviewer 2 Report

The new supplementary figure is hard to interpret without showing a kinetic trace outlining where the samples were collected (rather than referencing another paper).

It would have been ideal if the authors tested the binding of their peptide to different conformers rather than doing TEM, which is subjective to surface-mass transport issues during sample deposition and differential binding of Ab42 conformers to the TEM grid. I don't think their TEM alone adds much evidence as to which species their peptide actually binds. The paper would be strengthened if they can quantify the binding of their peptide to at least monomers and fibrils (and ideally also stabilized oligomers). I don't think this experiment is essential at this stage, but tempering the text to make clear this limitation would be good to ensure readers understand the exact species being bound by the peptide are not fully understood.

Author Response

This manuscript is a resubmission of an earlier submission. The following is a list of the peer review reports and author responses from that submission.

Round 1

Reviewer 1 Report

The current manuscript by Kim and colleagues suggests the development of low molecular amyloid beta 42 detection system using small peptide-based enzyme-linked peptide assay. Overall, I find the manuscript potentially interesting, especially for the field of Alzheimer’s disease biomarkers. Nevertheless, it has some major flaws that need to be corrected in the revised version:

  1. The manuscript is poorly written. The writing style is often confusing or redundant. I strongly recommend English proofreading from a professional or a native speaker.
  2. Authors should perform statistical comparisons between groups in all graph presented. This is of key importance for the reader to understand the relevance of the author’s results.
  3. Table 3: What is the STDV of the Average recovery (%)? How do the author explain a % of recovery above 100%?
  4. Table 4: All the acronyms should be listed in the legend.
  5. Overall, Figure legends are very incomplete. Often the statistical method used, number of replicates, level of significance, etc. are missing. I strongly suggest major improvements for the manuscript’s data presentation in the revised version.

Reviewer 2 Report

This article is devoted to a very interesting and relevant topic - measuring the concentration of different forms of beta-amyloid in CSF patients. This is a very difficult task that researchers from all over the world are solving. The authors proposed a very attractive idea - to create a test system based on a highly specific and high-affinity peptide. However, I have a number of questions and comments for the authors.

 - Give physiological standard concentrations of amyloid beta peptide in CSF from literature as a reference value.
 - In Fig. 2A shows a very weak binding constant. The drawing is badly signed.
- In Table 2, make a monospaced font for peptide sequences.
- Please, provide in the text for which Abeta epitopes were used antibodies?
- Specify in detail the method of preparation of the amyloid solution.
- Figure 7 raises many questions. It is not clear what is shown in the picture. Why do the authors talk about distinguishing AD+ and AD- patients if the values ​​overlap significantly? What is ΔOD450 in the pictures? Please sign. What are AD+ and AD-? Please, explain in the text under Figure 7.
- I would like to see the value of the concentration of peptide in CSF, which the authors measured for AD- and AD+ patients.
A more serious question concerns the sensitivity of the test system. According to my recalculation, the sensitivity of the proposed system is in the range from 150 nanomolar to 10 micromolar. According to our data, the concentration of the Abeta peptide in the CSF in patients is nanomolar. For effective testing, the constant Kd must be lower by at least two orders of magnitude.
In addition, binding specificity has not been proven. After removing the amyloid from the CFS, it is necessary to show by any available method that it was the amyloid that interacted with the peptides. For example, on immunoblotting, that only Abeta amyloid peptide sat down. Or supersensitive phoresis. You can paint the membrane for protein. Stain with antibodies for amyloid. As a control, you can take the reverse peptide, for example.
I would also like to note that it is much more convenient for patients to use blood plasma for diagnostics.
The last remark is controversial.

typos: 

Line 110: complexity of 1010

Line 164: (anti-Aβ40 and anti-amyloid beta protein) - please, explain. 

Reviewer 3 Report

The manuscript of Kim et al., is in general, a well written and well-organized article addressing a relevant issue of amyloid-β detection. It was shown that by mimicking the ELISA system for Aβ42 detection, it is possible to observe Aβ that could be relevant for the reason that fast and cheap Aβ detection systems are still need to be developed. 

General comment:

-The us of monomeric Abeta for diagnostic of AD is very controversial, a huge body of evidence exists demonstrating that monomeric Abeta is rather innocuous. Due to the low specificity of amyloid proteins for AD diagnostic (amyloid plaques could exist, for example, in cognitively healthy individuals), other biomarkers are currently in the main focus. The authors may read here: https://www.nature.com/articles/s41591-020-0755-1

-The Authors do not discuss existing, well-validated and clinically approved assays such as Aβ peptides' detection assays provided by Meso Scale Discovery and Simoa® Technology. Meso Scale Discovery and Simoa® Technology provide very sensitive and reproducible products to detect Aβ. Thus, in light of existing assays, the added value of the new product presented in the manuscript is not clear.

- I disagree with the statement, Lines 58-59: Moreover, antibodies have an obvious disadvantage in detecting Aβ42. Because Aβ42 is a very small protein (4kDa), the much larger antibody can detect only Aβ42 oligomer over a 59 certain size in a sandwich ELISA system [22, 23]

Authors may, for example, read Linse et al., 2020, doi: 10.1038/s41594-020-0505-6

In light of the statement about, what would be the difference for the anti-amyloid antibodies used in the study? Please provide the charactrization of the antibodies. How may FTIC conjugation affect antibody- peptide interaction?

- Biopanning it is a new interesing approach and therefore should be better presented.

- Reasons for neuropsychological assessment (in relation for the current study) should be discussed. The median age of 66 years it is quite advanced. Please provide a list of patients indicating Braak stages.

  Line 38: Authors should consider recent advances related to AD treatment (Aduhelm (aducanumab-avwa) is an amyloid beta-directed antibody indicated for treating Alzheimer’s disease).

 Line 40: References are not appropriate/outdated. Other fragments (N- and C- terminus truncated Abeta) of APP cleavage not discussed.

Round 2

Reviewer 1 Report

Although the authors partially addressed of my concerns, the overall quality of the manuscript, e.g. data presentation and analysis, remains below average. 

Reviewer 2 Report

According to Fig 7. there is no difference in Ab42 concentration between AD and non-AD patients, is there? 

Reviewer 3 Report

IThe mansucrip now looks more clear! I respect all efforts Authors made for revision. I have no major coments ledt. There are a few vague statements, that I suggest either romove or specify/ add references:

-there is still a long way to go to cure AD.

-Although the mechanism of action is unclear, .

-Furthermore, the use of peptide aptamers for the detection of Aβ42 has some advantages, especially for detecting low molecular weight Aβ42 oligomers, owing to their small size. (reference is missing, what is a small size+ ? amyloid oligomers in AD CSF can be as small as  3 x 50 nm, De S at al., 2019 10.1186/s40478-019-0777-4)

-I have never seen an  expression authors used in their responce:  *Please do need full*, my I appologies if I missed something.